# Effects of unilateral posterior missing-teeth on the temporomandibular joint and the alignment of cervical atlas

**Tsun-Hung Fang[1], Meng-Ta Chiang[1,2], Ming-Chun Hsieh[1], Ling-Yu Kung[1], Kuo-Chou Chiu**[1,2]*

**1** Department of Family Dentistry, Tri-Service General Hospital, National Defense Medical Center, Taipei, Taiwan, Republic of China, **2** School of Dentistry, National Defense Medical Center, Taipei, Taiwan, Republic of China

* scalingdentist61@yahoo.com.tw

**Data Availability Statement:** All relevant data are within the manuscript.

**Funding:** This project was funded by Southern Taiwan Science Park Bureau,Taiwan, grant

## Abstract

Cervical atlas alignment changes are associated with craniofacial development. Disturbance of craniofacial development may be associated with temporal mandibular joint function. Therefore, we examined the possibility of a correlation between unilateral missing teeth and morphologic changes of the spine and posture. We collected eighty-nine patients (38 men and 51 women) with unilateral posterior missing teeth and twenty patients without previous orthodontic treatment or missing posterior teeth by tracing and analyzing their panoramic and cephalometric film. We measured the angulations of articular eminence, craniocervical angle, and the percentage of the occlusal plane passing through the first and second cervical vertebrae with other morphologic geometric data. The angle of articular eminence inclination was higher in the non-missing teeth group than the missing teeth group (46.66˚ and 42.28˚, respectively). The cranio-cervical angle was smaller in the missing posterior teeth group than the non-missing posterior teeth group (99.81˚ and 103.27˚, respectively). The missing teeth group also showed fewer occlusal planes passing through the intersection of the first and second cervical vertebrae compared to the non-missing teeth group (28.9% and 65%, respectively). Individuals with unilateral missing teeth had lower articular eminence inclination, smaller cranio-cervical angle, and a lower percentage of the occlusal plane passing through the intersection of the first and second cervical vertebrae.

## Introduction

In the human body, the occlusion, craniosacral system, and functional systems of the body are profoundly intertwined in a dynamic balance. The temporal mandibular joint (TMJ) is one of the most complex joints in the human body [1, 2]. It is a synovial sliding-ginglymoid joint which is harmoniously controlled by a complex neural-muscular system [3]. Anatomically, the temporomandibular articulation is located on the undersurface of the squamous part of the temporal bone: the glenoid fossa. The articular disk separates the glenoid fossa into the superior and inferior joint cavities. The mandible condyle, located in the inferior joint cavity, may

number: BX-06-09-14-107, Tri-Service-General Hospital, Taiwan, grant number TSGH-C107-018, TSGH-C107-146, and Ministry of Science and Technology, Taiwan, grant numbers Most 107-2314-B-016-047.

**Competing interests:** NO authors have competing interests.

achieve mouth opening and closing, protrusion and retrusion, alternate lateral movement, and provide the balance of the jaw [4]. The slope of the articular eminence (AE) adjusts the pathway of the TMJ. Any transforms in dental anatomy and mechanics will relatively alter the craniosacral system and other functional systems in the body [2, 4].

The AE inclination may change following tooth wear and tooth loss. It may expedite the degeneration of the TMJ component [5, 6]. Our previous study also revealed decreased AE inclination angulation in the missing posterior teeth side compared with the non-missing posterior teeth side [7]. The missed single posterior teeth may advance the bone remodeling of the ipsilateral AE with more bone resorption.

The changing of the AE is associated with TMJ problems [7]. Cranio-cervical posture is a crucial factor in craniofacial architecture and TMJ dysfunction [8]. The cranio-cervical angle was adopted to evaluate the advancement of the mandible. An increase in the cranio-cervical angulation is associated with the reduced mobility of the TMJ [9]. Research has revealed cranio-cervical postures, such as cranio-cervical angulation and mal-posturing of the first cervical vertebra (C1) and second cervical vertebra (C2), may be associated with the TMJ dysfunction. [10]. C1 located at the occipital condyle's joint surface, joining the skull at the atlanto-occipital joint. It suggests an indispensable role for jaw mechanics in the development of head posture. Disturbing the posture of C1 and C2 are informally akin to spinal and head posture abnormalities; neurological well-being may follow with TMJ dysfunction [9–11]

The study's purpose was to examine the relationships between the unilateral missing posterior tooth and the inclination of the articular eminence, or the atlas alignment using a straightforward, uncomplicated approach, low-radiologic dose risk, and low-cost radiology research method, by providing an easily obtained measured values such as unilateral tooth loss and inclination changes values in atlas or AE, that help dentists to be aware of potential temporomandibular problems.

## Material and methods

### Population

This study was approved by the National Defense Medical Center Research & Ethics Committee. The participants recruited in this study were patients of the Department of Dentistry, Tri-Service General Hospital, Taipei, Taiwan. All images investigated in this study were retrieved from an encrypted, confidentially protected, dental X-ray database. The measurements performed retrospectively analyzing panoramic and cephalometric film records of 178 Joints from 89 patients with unilateral one or more than one posterior tooth loss, who had previously visited the clinic as the edentulous group. Those patients ages ranged from 18 to 69 years. Another forty joints collected from twenty patients taking the panoramic film during annual dental follow up (ten males and ten females) without any missing teeth or orthodontic treatment as the control group. The control group ages ranged from 20 to 69 years. The exclusion criteria were as follows:

1. The presence of congenital craniofacial abnormalities. Any systemic diseases which may affect joint morphology: rheumatoid arthritis, hemihyperplasia, hemifacial atrophy, Paget's disease, and any cervical spine diseases.

2. Patients with fractures or pathologic lesions in the region of the AE interfered with the measurement of the region.

3. Patients who had prosthetic reconstruction with crown and bridge or implants on missing teeth site.

4. Patients who received previous orthodontic treatment.

## Imaging procedures and measurements (panoramic radiography and cephalometric radiography)

A single operator performed all the radiographic images using the same radiographic unit (CRANES EXCEL CEPH, SORDEX, Milwaukee, WI, USA) with exposure factors of 67 kVp and 10 mA. The Pangea Dental (EBM Technologies, Taipei, Taiwan) software program was applied in statistical analysis. The images were shown on a 100% scale LCD monitor. Each measurement was repeated twice by two examiners. Two observers (T.F. Fang & M.T. Chiang) traced the panoramic and cephalometric radiographs independently to coordinate their findings before making interpretations.

In the panoramic analysis, the sagittal outlines of the left and right AE and glenoid fossa were traced under the monitor. Bilateral "orbitale" (the lowest point in the margin of the orbit) and the "porion" (the highest point in the margin of the auditory meatus) were identified. By joining the orbitale and porion, we confirmed the Frankfurt horizontal plane, and then identified the most superior point on the glenoid fossa (the crest of glenoid fossa) and the most inferior point on the AE (the crest of AE). By joining the crest of the glenoid fossa and the crest of AE, we identified the mean condylar path inclination (CPI) The AE inclination was measured using the top-roof line method, which was the angle between the CPI plane and the Frankfort horizontal plane (Fig 1) [3, 12].

The cranio-cervical angle (CCA) measurement follows Rocabado's description of the cephalometric analyses published in 1984 [8]. The CCA is an angle with the intersection of the McGregor Plane (tangent from the base of the occipital bone until it reaches the posterior nasal spine on the hard palate) and the Odontoidium Plane (starting from the apex of the odontoid process of C2 to the most anterior and inferior point of the body of C2) [13]. We use this angle to evaluate the anterior-posterior position of the cranium with the cervical spine.

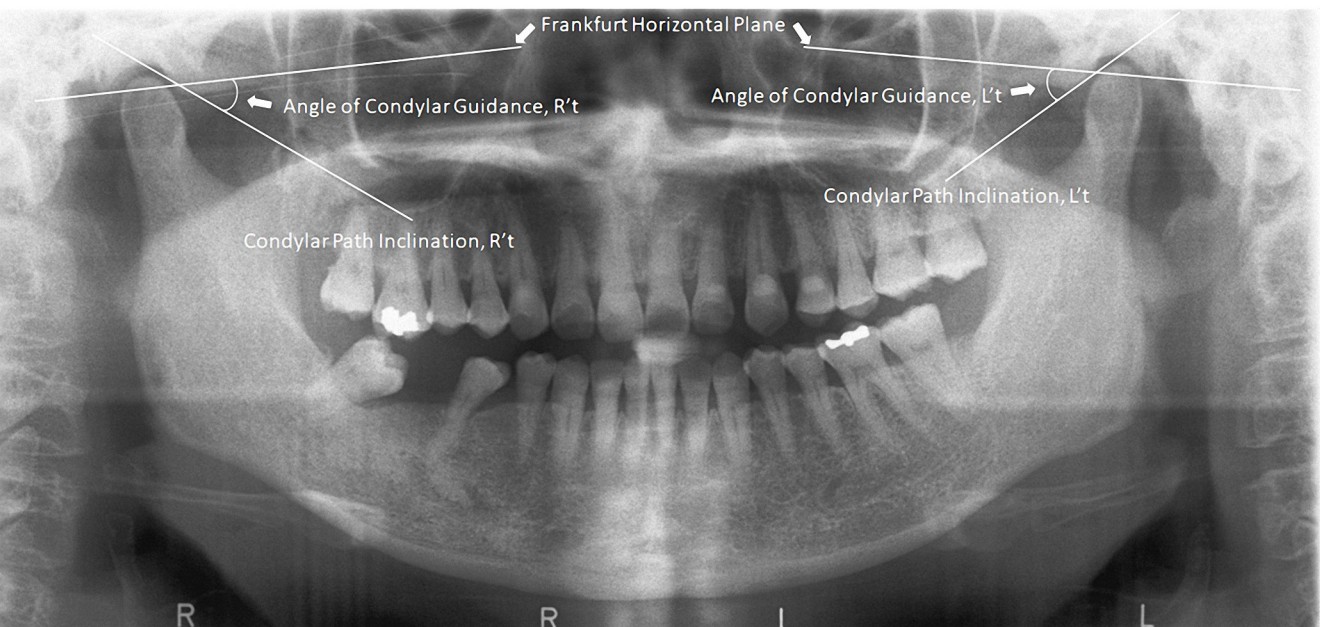

**Fig 1. Panoramic radiograph tracing.** Panoramic radiograph of the patient showing the trace of the angle of sagittal condylar guidance.

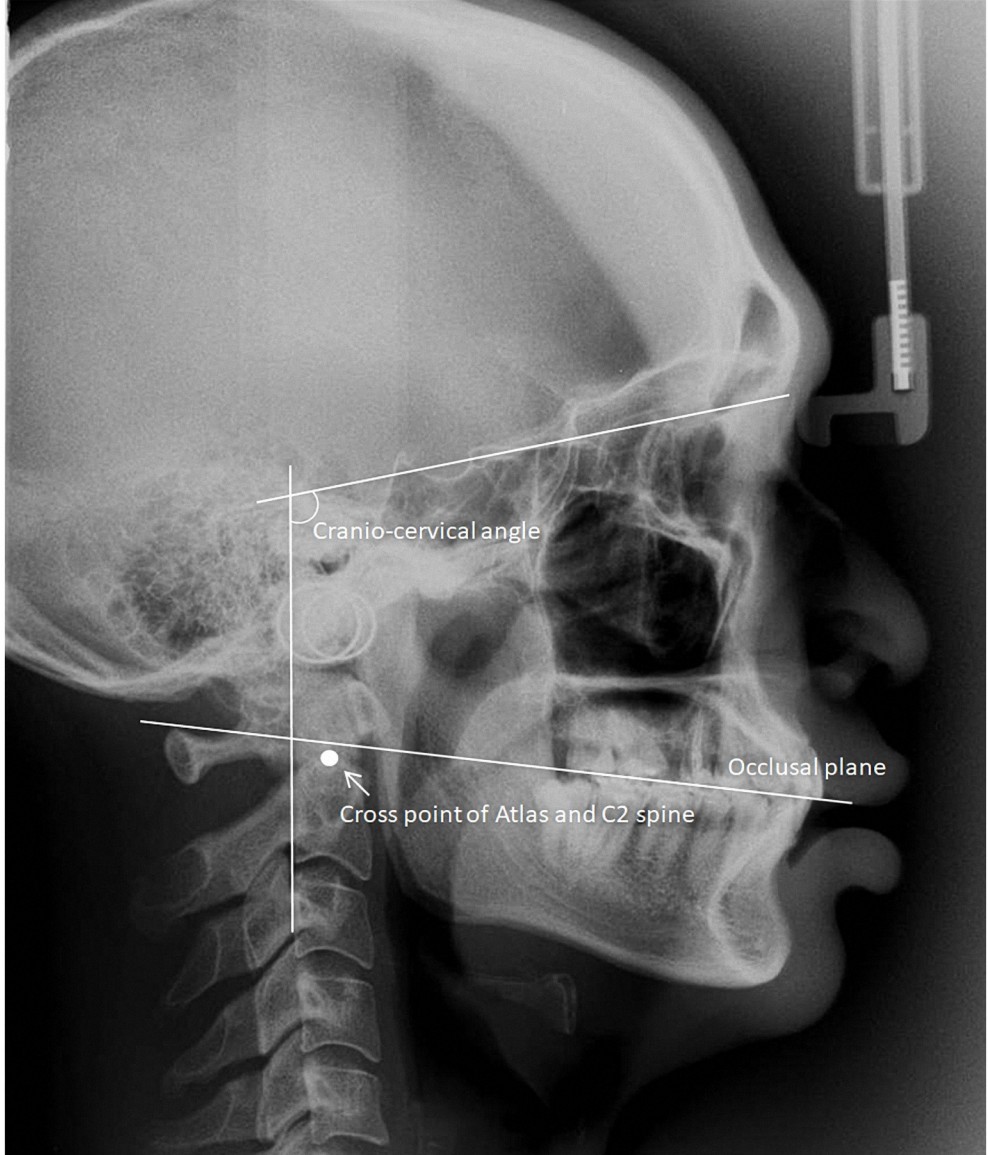

**Fig 2. Cephalometric radiograph tracing.** Cephalometric radiograph of the patient showing the trace of the cranio-cervical angles, the extension line of the upper occlusal plane, and the cross point of the atlas and the second cervical vertebra.

Values between 96˚ and 106˚ were considered normal. Values lower than 96˚ suggest an extension of the head; values higher than 106˚ indicate a flexed position of the head, causing it to be in an anterior position [14]. We also traced the relationships between the extension line from the occlusal plane (OP) and the cross point between the atlas and C2 (OP to C1-C2) from cephalometric radiography. The idea occlusion is definite as the occlusal plane passing through the cross point of C1-C2 (Fig 2) [15].

## Statistical analysis

We used SPSS 24.0 software (Statistical Package for the Social Sciences, IBM Software Group, Armonk, NY, USA) for all the statistical analyses. The independent t-test was applied at the

**Table 1. Measured AE inclination angles of missing teeth with non-missing teeth in panoramic radiography.**

|  | No. | Average Angles (˚) | S.D. | T | *P*-value[a] |
|---|---|---|---|---|---|
| **Missing side** | 89 | 42.28 | 8.87 | -7.54 | <0.001 |
| **Non-missing side** | 89 | 46.66 | 9.02 |  |  |

[a]*p* value was calculated using the Paired t-test.

descriptive statistics level to evaluate the differences between arithmetic means with significance. We applied the Chi-square test to evaluate the percentage of equality and the mean absolute difference, to show the asymmetry of the measured inclination of the unilateral missing and non-missing side joint, on the same image. The paired t-test was used to determine differences in the bilateral, mean, and AE inclination of the experimental group. We also applied the linear regression and logistic regression model to evaluate the influence of age change on the cephalometric analysis values between the control group and the experimental group. A P value of $< 0.05$ was considered statistically significant.

## Results

There were 38 males and 51 females included in this study. The average age was 36 years old; the range was 20 to 69 years old.

The measured AE inclination angles of the unilateral missing teeth side and the non-missing teeth side were shown in Table 1. The mean measured value for the AE inclination in missing teeth side was 42.28˚, with a standard deviation of 8.87˚; the non-missing teeth side was 46.66˚, with a standard deviation of 9.02˚. The AE inclination of the non-missing teeth side was slightly steeper than that of the missing teeth side and considered statistically significant (P <0.001, Table 1).

There is no gender difference in the missing teeth group and the non-missing teeth group. The missing teeth group had older mean age, compared to the non-missing teeth group. The cranio-cervical angle in the non-missing teeth group is 103.27˚ compared to 99.81˚ in the missing teeth group, with a statistical significance (P<0.05, Table 2). In the missing teeth

**Table 2. Factors related to unilateral posterior missing teeth.**

| Variable | Non-Missing teeth n = 20 | Missing teeth n = 89 | *P*-value |
|---|---|---|---|
| **Gender** |  |  | 0.552 |
| Female | 10 (50%) | 51 (57.3%) |  |
| Male | 10 (50%) | 38 (42.7%) |  |
| **Age**, mean (SD) | 22.3 (1.84) | 36.09 (12.83) | <0.001 |
| **Cranio-cervical angle**, mean (SD) | 103.27 (5.08) | 99.81 (8.7) | 0.022 |
| **Does the occlusal plan pass through the intersection of first and second cervical vertebrae?** |  |  | 0.002 |
| No | 7 (35%) | 64 (71.91%) |  |
| Yes | 13 (65%) | 25 (28.09%) |  |

※ mean (SD) with independent t test; n (%) with Chi-square test

**Table 3. Changing differences between missing teeth and non-missing teeth adjusted by age.**

| Variable | B (SE) | P-value[a] | 95% CI |
|---|---|---|---|
| **Dependent variable: Cranio-cervical angle** | | | |
| **Group** (Missing teeth vs. non-missing teeth) | -5.825 (2.172) | 0.008 | (-10.13, -1.519) |
| **Age** | 0.172 (0.066) | 0.011 | (0.041, 0.303) |
| Variable | OR | P-value[b] | 95% CI for OR |
| **Dependent variable:** | | | |
| Does the occlusal plane pass through the intersection of first and second cervical vertebrae?(0:No; 1:Yes) | | | |
| **Group** (Missing teeth vs. non-missing teeth) | 0.302 | 0.038 | (0.098, 0.936) |
| **Age** | 0.972 | 0.16 | (0.934, 1.011) |

[a]$p$ values were calculated using the Linear Regression.

[b]$p$ values were calculated using the Logistic Regression.

group, only 28.09% showed the occlusal plane passing through the intersection of the first and second cervical vertebrae (Table 2).

Linear regression and logistic regression evaluations showed a statistically significant lower cranio-cervical angle in missing teeth group compared with the non-missing teeth group, after adjusted for age (P<0.05, Table 3). Significantly less occlusal planes pass through the intersection of the first and second cervical vertebrae in missing teeth group (P = 0.038, OR = 0.302, Table 3).

## Discussion

Anatomically, the atlas is the most superior cervical vertebrae of the spine. It supports the globe of the head and was named after the Atlas of Greek mythology. The atlas is the top-most vertebrae connecting the skull and spine via a joint. The atlas and axis bore more range of motion than normal vertebrae [11]. On the vertebrae column, the alanto-occipital joint allows the head to nod up and down with head movement.

The dens act as a pivot, allowing the atlas, attached to the head, to rotate on the axis. Correlations between facial morphology and cranio-cervical angle show that with changes of the cranio-cervical angle, the atlas may follow with the cervical column [8, 14].

The cranium's anterior-posterior position to the cervical spine was analyzed with the cranio-cervical angle. Values between 96˚ and 106˚ are considered normal. Values less than 96˚ are considered an extension of the head; values higher than 106˚ indicate a flexed position of the head, as an anterior position [14]. Although our control and study groups were under average values, the missing-teeth group showed statistically significant lower angulations than those of the non-missing teeth group. On the facial skeleton area, the morphology differences with the changing of the cranio-cervical angle may associate with the facial muscle forces [16]. When the head is extended to the cervical column, forces on the bone structures increase; the layer would passively elongate, and the forces would limit both maxillary growth and mandible forward growth [9, 16].

The Quadrant Theorem suggested that the apex of the combined muscular control of the mandible, in all functioning movements, located at the dens between the atlas and axis [17]. At resting position, muscle controlling the pivotal axis of the mandible occurs at the dens, between the atlas and axis [18]. The balance of muscle tension with TMJ may reveal by tracing the occlusal plane to determine if it crosses the C1 and C2 intersection [17, 18]. As a result, if the mandibular jaw has misalignment or dysfunction, patients will have an awkward position of C1-C2; the neurological well-being and unbalance of muscle tension will cause

temporomandibular joint dysfunction (TMD) [4, 19]. Our data showed that the missing-teeth group has a higher occurrence of an awkward position of C1-C2, with a much higher chance of mandible jaw misalignment or dysfunction, according to the Quadrant Theorem [19].

Head posture is highly associated with cranio-facial morphology changes. The AE inclination change influences subsequent craniofacial growth [20, 21]. For example, obstruction of the upper airway may lead to postural change, resulting in the extension of the cranio-cervical angle [22]. The soft-tissue stretching will trigger differential forces on the skeleton [23]. These cyclic changes may influence TMJ dysfunction. Research shows that minutes after physiologically balanced molar support, the vertical head, shoulder, spine, and pelvic posture begins to normalize [24]. Without one-sided posterior tooth support, the AE compensatory mechanisms remodel the missing section with time [5]. AE changes may be caused by overuse of one side of the dental arches during mastication, resulting in an uneven allocation of biomechanical forces [25, 26]. With aging and tooth loss, AE remodeling and biomechanical conditions may affect soft-tissue stretching, and disturb the balance of mandibular muscle control [5, 14, 25, 27].

This study is the first to explore the relationship between missing teeth, AE inclination change, and cervical posture. These changing relationships may be associated with TMJ development. Our non-missing teeth group included young college students; an age bias may influence the results. Increasing the population database may show more strength with our findings. The distortion related to the panoramic film may influence the result of the angulation measurement. Correctly positioning patients in fully extend the cervical spine and confirm patients' chin was on the chin rest with checking patients' ala-tragus line is approximately horizontal can eliminate the distortion from technic.

## Conclusion

Our study showed that people with single missing teeth might have a decreased AE inclination angle. The unipartite missing teeth may decrease the cranio-cervical angle and cause more deviation of the occlusal plane from the C1-C2 intersection. It causes disturbing posturing of C1 and C2. These changes might influence the balance of TMJ biomechanical or physiologic development.

## Supporting information

**S1 File.**
(XLSX)

## Author Contributions

**Conceptualization:** Meng-Ta Chiang, Kuo-Chou Chiu.

**Data curation:** Tsun-Hung Fang, Ming-Chun Hsieh.

**Investigation:** Tsun-Hung Fang, Meng-Ta Chiang, Ling-Yu Kung.

**Methodology:** Meng-Ta Chiang.

**Project administration:** Ling-Yu Kung, Kuo-Chou Chiu.

**Validation:** Ming-Chun Hsieh.

**Writing – original draft:** Tsun-Hung Fang.

**Writing – review & editing:** Kuo-Chou Chiu.

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
