## [Decision Letter · Decision Letter 0]

18 Sep 2020

PONE-D-20-23912

Effects of unipartite posterior missing-teeth on the temporomandibular joint and the alignment of cervical atlas

PLOS ONE

Dear Dr. CHIU,

Thank you for submitting your manuscript to PLOS ONE. After careful consideration, we feel that it has merit but does not fully meet PLOS ONE’s publication criteria as it currently stands. Therefore, we invite you to submit a revised version of the manuscript that addresses the points raised during the review process.

We look forward to receiving your revised manuscript.

Kind regards,

Essam Al-Moraissi

Academic Editor

PLOS ONE

Journal Requirements:

2. Please ensure that you describe in your methods section how capacity to consent was determined for the participants in this study.

Reviewers' comments:

Reviewer's Responses to Questions

**Comments to the Author**

1. Is the manuscript technically sound, and do the data support the conclusions?

Reviewer #1: Yes

Reviewer #2: Yes

Reviewer #3: Yes

2. Has the statistical analysis been performed appropriately and rigorously? 

Reviewer #1: I Don't Know

Reviewer #2: Yes

Reviewer #3: Yes

3. Have the authors made all data underlying the findings in their manuscript fully available?

Reviewer #1: Yes

Reviewer #2: Yes

Reviewer #3: Yes

4. Is the manuscript presented in an intelligible fashion and written in standard English?

Reviewer #1: Yes

Reviewer #2: Yes

Reviewer #3: Yes

5. Review Comments to the Author

Reviewer #1: Title

The term “Unipartite” used to describe unilateral missing molars undoes not appear in the literature (trending)  and confuses the reader. As well it may accurately indicate a missing tooth but not whether there was missing posterior support unilaterally or not.

Could you please see my attached review, I have added all my comments to the Author in an attachment.

Reviewer #2: Good study. It would have been better to take CBCT rather than panoramic and lateral cephalogram for the study subjects. Which would give much better angulation and metrics. Also in some cases the ambulation might also change due already underlying inflammatory changes.

Reviewer #3: In this paper, the authors studied the correlation between unipartite missing teeth and morphologic changes of the spine and posture. Angulations of articular eminence, cranio-cervical angle, and the percentage of the occlusal plane passing through the first and second cervical vertebrae with other morphologic geometric data were measured. It is novel and of clinical interest.

It is novel and has clinical significance.

It is known that head posture is highly correlated with craniofacial changes. On the other hand, head posture also affects the above-mentioned angle measurement. In fact, the standard procedures of radiography are sufficient for clinical diagnosis. However, in this study, it is uncertain whether measurement errors (including the reproducibility of the subject's head posture) will affect the results. In particular, it is recommended to verify the measurement error of the current scheme while maintaining sufficient reproducibility of the posture.

6. PLOS authors have the option to publish the peer review history of their article (what does this mean?). If published, this will include your full peer review and any attached files.

Reviewer #1: **Yes: **Dr. Curtis Westersund

Reviewer #2: No

Reviewer #3: No

---

## [Author Response · Author response to Decision Letter 0]

23 Oct 2020

Reviewer #1: I have incorporated your suggestions into my revision. They were constructive and valuable for improving our research. Thank you.

Reviewer #2: I have incorporated your suggestions into my revision. They were constructive and valuable for improving our research. Thank you for your comments.

Reviewer #3:I have incorporated your suggestions into my revision. Thank you for your help.

---

## [Decision Letter · Decision Letter 1]

9 Nov 2020

Effects of unilateral posterior missing-teeth on the temporomandibular joint and the alignment of cervical atlas

PONE-D-20-23912R1

Dear Dr. CHIU,

We’re pleased to inform you that your manuscript has been judged scientifically suitable for publication and will be formally accepted for publication once it meets all outstanding technical requirements.

Kind regards,

Essam Al-Moraissi

Academic Editor

PLOS ONE

Additional Editor Comments (optional):

Reviewers' comments:

Reviewer's Responses to Questions

**Comments to the Author**

1. If the authors have adequately addressed your comments raised in a previous round of review and you feel that this manuscript is now acceptable for publication, you may indicate that here to bypass the “Comments to the Author” section, enter your conflict of interest statement in the “Confidential to Editor” section, and submit your "Accept" recommendation.

Reviewer #1: All comments have been addressed

Reviewer #2: All comments have been addressed

Reviewer #3: All comments have been addressed

2. Is the manuscript technically sound, and do the data support the conclusions?

Reviewer #1: Yes

Reviewer #2: Yes

Reviewer #3: Yes

3. Has the statistical analysis been performed appropriately and rigorously? 

Reviewer #1: I Don't Know

Reviewer #2: Yes

Reviewer #3: Yes

4. Have the authors made all data underlying the findings in their manuscript fully available?

Reviewer #1: Yes

Reviewer #2: Yes

Reviewer #3: Yes

5. Is the manuscript presented in an intelligible fashion and written in standard English?

Reviewer #1: Yes

Reviewer #2: Yes

Reviewer #3: Yes

6. Review Comments to the Author

Reviewer #1: I found 2 errors in formatting:

"Pdontoideum Plane" is likely meant to read Odontoideum Plane in the first paragraph on page 6

Not sure what this means at the bottom of page 9 - "joint11" but I think they have a misplace reference.

Reviewer #2: Article is good for publishing. It's a nice research and an original. The study attempts the readers to a very specific approach for diagnosis of Internal derangement

Reviewer #3: The authors did minor revision on the manuscript such as usage of terms, grammatical error etc. The manuscript is nice in the current form for publication.

7. PLOS authors have the option to publish the peer review history of their article (what does this mean?). If published, this will include your full peer review and any attached files.

Reviewer #1: **Yes: **Curtis Westersund

Reviewer #2: No

Reviewer #3: No

---

## [Editor Report · Acceptance letter]

19 Nov 2020

PONE-D-20-23912R1 

Effects of unilateral posterior missing-teeth on the temporomandibular joint and the alignment of cervical atlas 

Dear Dr. Chiu:

I'm pleased to inform you that your manuscript has been deemed suitable for publication in PLOS ONE. Congratulations! Your manuscript is now with our production department. 

Kind regards, 

on behalf of

Dr. Essam Al-Moraissi 

Academic Editor

PLOS ONE